# DeepSpaceYoloDataset: Annotated Astronomical Images Captured with Smart Telescopes

Olivier Parisot

Luxembourg Institute of Science and Technology, 5 Avenue des Hauts-Fourneaux,
L-4362 Esch-sur-Alzette, Luxembourg; olivier.parisot@list.lu

**Abstract:** Recent smart telescopes allow the automatic collection of a large quantity of data for specific portions of the night sky—with the goal of capturing images of deep sky objects (nebula, galaxies, globular clusters). Nevertheless, human verification is still required afterwards to check whether celestial targets are effectively visible in the images produced by these instruments. Depending on the magnitude of deep sky objects, the observation conditions and the cumulative time of data acquisition, it is possible that only stars are present in the images. In addition, unfavorable external conditions (light pollution, bright moon, etc.) can make capture difficult. In this paper, we describe DeepSpaceYoloDataset, a set of 4696 RGB astronomical images captured by two smart telescopes and annotated with the positions of deep sky objects that are effectively in the images. This dataset can be used to train detection models on this type of image, enabling the better control of the duration of capture sessions, but also to detect unexpected celestial events such as supernova.

**Keywords:** YOLO; electronically assisted astronomy; smart telescope

## 1. Summary

Smart telescopes are automated devices combining optical components, specialized cameras, and tracking mounts to capture the images of deep-sky objects (DSOs) like galaxies, nebulae, and even star clusters. Through tablets or smartphones, they enable admiring faint celestial targets that are invisible through direct observation: sensors are much more sensitive than the human eye and are able to capture faint signals coming from deep space, and lightweight on-board processing combines can capture short-exposure images to produce high-quality final images [1]. Smart telescopes are dedicated to electronically assisted astronomy (EAA), allowing stargazing sessions with family and friends even during cold nights [2]. Furthermore, they are effective in the observation of the night sky in geographical areas impacted by light pollution present in cities and suburban areas—where only a few targets are visible with direct observation [3].

Smart telescopes are also of obvious scientific interest. In fact, the more cumulative time that camera with a telescope spends capturing images for a portion of the sky, the higher the signal-to-noise ratio of the resulting data. Many astronomical objects like galaxies or nebula emit faint light, and by collecting light during an extended period, a camera with a telescope can gather more photons, improving the signal-to-noise ratio [4]. This makes it easier to distinguish the faint signal of the astronomical object from the background noise. For example, an important discovery was made during an amateur/professional collaboration using backyard material: an unknown cloud near the Andromeda galaxy (Messier 31) was finally observed on images, by accumulating data during a long period of time [5]. Furthermore, the simultaneous use of networks of smart telescopes can contribute

to the study of asteroids and even exoplanets [6], especially to observe a larger field of view with an increased resolution—allowing for finer details to be discerned and confirming the consistency of observed features.

Smart telescopes allow for the precise programming of the starting and stopping capture of images of a specific part of the starry sky, and the results are recorded on a storage medium for later use (such as a portable hard drive or cloud servers). This is where a very important feature can be of great use: the detection of objects that are effectively visible in the captured images. While there is little doubt about the presence of stars in the images, it is more difficult to be certain of having captured a galaxy or nebula in the image, particularly when aiming at difficult high-magnitude targets that require long hours of capture. In addition, unfavorable outdoor conditions like light pollution or a bright moon can make it difficult to obtain images of sufficiently high quality. It is therefore useful to have a detection model to automatically analyze images and produce annotations with the detected objects. In practice, such a detection model would make it possible to determine whether additional data acquisition by the telescope is required to effectively capture the desired targets. It could also be used to detect whether unknown and unexpected celestial events have occurred—as was the case very recently with supernova SN 2023ixf, which appeared in galaxy M101 [7].

In this work, we describe DeepSpaceYoloDataset, a set of astronomical images captured with smart telescopes and then annotated to indicate the position of visible DSO.

The rest of this paper is organized as follows. Firstly, related works about astronomical object detection methods and data are discussed (Section 2). Secondly, the dataset is described (Section 3). Thirdly, the method to build the dataset is detailed (Section 4). Finally, a data usage example is presented (Section 5), and we conclude by opening some perspectives (Section 6).

## 2. Related Works

Traditionally, astronomical object detection is realized by using astrometry (i.e., by finding the exact position, scale, and orientation of the image): by mapping the result on well-known night sky maps (containing exact DSO positions), it is then possible to find what objects are visible on the analyzed image [8]. In fact, simplified astrometry/plate solving is used by smart telescope for the automated initialization. It is effective, but one of the drawbacks is that it requires access to a database containing sky maps, or a network access to communicate with online services. It does not help to detect unexpected events like supernova [7].

Computer vision (CV) approaches for object detection are also numerous, as they enable the direct extraction of information from images. Ten years ago, an interesting work based on image segmentation was proposed to detect both galaxies and stars in large survey images [9]. Recently, several techniques based on artificial intelligence (AI) have been suggested. Among them, state-of-the-art YOLO-based approaches (You Only Look Once) were especially dedicated to object detection in images. For example, [10] proposes to combine an improved YOLOv2 model and data augmentation to detect and classify galaxies in large astronomical surveys. [11] applies a customized version of YOLOv7 to detect space debris using a dedicated training set [1].

The annotated datasets of astronomical images were already published, but they are not efficient to train a detection model for images captured with smart telescopes.

- AI-based methods need huge training data to be effective. For instance, megacosm is a set of annotated pictures with the positions of various celestial bodies—but it contains insufficient images (400) [12].
- Astronomical images are noisy, and methods like YOLO are sensitive to noise and need to be trained on realistic datasets [13]. To build an effective training set, using high-quality images such as those obtained with Hubble and/or the James Webb Space Telescope is not relevant, and adding artificial noise to these images is not effective either because it will not be as realistic as real noise.

- Light pollution has an important and negative impact on the quality of astronomical images [3]: like for noise, it is important to have a training set reflecting this issue in images.
- Due to the lack of publicly available data, the PixInsight company recently launched the Multiscale All-Sky Reference Survey (MARS), an initiative to collect images from amateur astronomers in order to improve their own algorithms [2].

To the best of our knowledge, there is currently no annotated dataset based on the images captured from geographical zones impacted by light pollution and with equipment accessible to amateurs.

## 3. Data Description

DeepSpaceYoloDataset is a single-class dataset formatted with the YOLO standard, i.e., with separated files for images and annotations, usable by state-of-the-art training tools [14] and graphical software like MakeSense (1.11.0-alpha) [15]. More precisely, it is a ZIP file containing 4696 RGB images in JPEG format (minimal compression), and 4696 text files containing the positions of DSO. RGB images were collected with two smart telescopes between March 2022 and September 2023, from Luxembourg, France, and Belgium—i.e., in locations strongly impacted by light pollution (for more details, see Section 4). Each RGB image has a resolution of 608 × 608 pixels and corresponds to the capture of different zones of the night sky visible in Northern Hemisphere. DSO are identified in well-known astronomical catalogues (Table 1) and described in many books and sky map software [16].

**Table 1.** Identifiers of the DSO present in DeepSpaceYoloDataset images are sorted by catalogues. As DSO can be listed in different catalogues, we used this order: Messier, New General Catalogue, Index Catalogue, Sharpless, Abell.

| Catalogue | List of Targets |
|---|---|
| Messier | M1, M10, M100, M101, M102, M103, M104, M105, M106, M107, M108, M109, M11, M110, M12, M13, M14, M15, M16, M17, M18, M19, M2, M20, M21, M22, M23, M24, M25, M26, M27, M29, M3, M31, M33, M34, M35, M36, M37, M38, M39, M4, M41, M42, M44, M45, M46, M47, M48, M49, M5, M50, M51, M52, M53, M56, M57, M58, M59, M61, M62, M63, M64, M65, M67, M68, M71, M72, M74, M76, M77, M78, M8, M80, M81, M82, M83, M85, M86, M87, M9, M92, M94, M95, M96, M97, |
| New General Catalogue | NGC1023, NGC1027, NGC1055, NGC1245, NGC1275, NGC1333, NGC1342, NGC147, NGC1491, NGC1499, NGC1502, NGC1579, NGC1746, NGC1788, NGC185, NGC188, NGC1909, NGC1931, NGC1961, NGC1977, NGC2022, NGC2024, NGC2169, NGC2170, NGC2174, NGC2244, NGC225, NGC2261, NGC2264, NGC2282, NGC2359, NGC2360, NGC2371, NGC2392, NGC2403, NGC2419, NGC2420, NGC246, NGC2506, NGC2539, NGC2683, NGC281, NGC2841, NGC2903, NGC2946, NGC3077, NGC3115, NGC3190, NGC3344, NGC3628, NGC40, NGC4038, NGC4244, NGC4314, NGC4395, NGC4490, NGC4535, NGC4559, NGC4565, NGC457, NGC4631, NGC488, NGC4889, NGC5466, NGC5566, NGC559, NGC5907, NGC6144, NGC6210, NGC6229, NGC6342, NGC6537, NGC654, NGC6543, NGC663, NGC6633, NGC672, NGC6760, NGC6781, NGC6822, NGC6823, NGC6826, NGC6883, NGC6888, NGC6891, NGC6894, NGC6905, NGC6914, NGC6928, NGC6934, NGC6946, NGC6960, NGC6979, NGC6992, NGC7000, NGC7006, NGC7008, NGC7009, NGC7023, NGC7048, NGC7129, NGC7209, NGC7217, NGC7293, NGC7318, NGC7331, NGC7380, NGC7479, NGC752, NGC7606, NGC7635, NGC7640, NGC7662, NGC772, NGC7789, NGC7814, NGC7822, NGC864, NGC877, NGC884, NGC891, NGC925 |

**Table 1.** *Cont.*

| Catalogue | List of Targets |
|---|---|
| Index Catalogue | IC10, IC1318, IC1396, IC1795, IC1805, IC1848, IC2177, IC342, IC348, IC405, IC410, IC417, IC434, IC443, IC4592, IC4756, IC4955, IC5070, IC5146, IC59 |
| Sharpless | Sh2-101, Sh2-129, Sh2-155, Sh2-188, Sh2-216 |
| Abell | Abell24, Abell39 |

For each image, there is a text file with the same name, containing the positions of visible DSO in YOLO format:

```
class, x_center, y_center, width, height
```

Let us take an example with the first image and its associated label file (1.txt): two bounding boxes are defined for this image, namely one per row:

```
0 0.8026 0.7097 0.0690 0.0773
0 0.9925 0.4424 0.0148 0.0394
```

## 4. Methods

### 4.1. Image Acquisition

The images were captured between March 2022 and September 2023 from Luxembourg, France, and Belgium—by using the built-in alignment/stacking features of these two smart instruments:

- The Stellina smart telescope [3] is based on an ED doublet with an aperture of 80 mm and a focal length of 400 mm (focal ratio of f/5). It is equipped with a Sony IMX178 CMOS sensor with a resolution of 6.4 million pixels (3096 × 2080 pixels) (Figure 1).
- The Vespera smart telescope [4] is built on an apochromatic quadruplet with an aperture of 50 mm and a focal length of 200 mm (focal ratio of f/4). It is equipped with a Sony IMX462 CMOS sensor with a resolution of 2 million pixels (1920 × 1080 pixels).

The default parameters of Stellina and Vespera were applied: 10 s for exposure time and 20 dB for gain. This configuration is optimal: this exposure time is a good trade-off to obtain good images with the alt-azimuth motorized mounts of the instruments (which may give a higher value to undesired star trails).

For each observation session, the instruments were installed in a dark environment (no direct light) and properly balanced using a bubble level on a stable floor. Depending of the conditions and of the targets, city light suppression (CLS) or dual band filters were used to capture more data (especially for nebula).

RGB images were obtained with reasonable cumulative integration times (from 20 to 120 min—usually sufficient to obtain a good signal-to-noise ratio for most of the targets): in astronomy, the cumulative integration time refers to the total time during which observational data have been captured and effectively processed to obtain a final image of an object or region of the sky.

In total, 12,512 high-resolutions JPEG images were obtained, and we cropped them to 4696 patches of 608 ×608 pixels in order to match to the default YOLO resolution [17].

Given that we wanted data obtained from an area heavily impacted by light pollution (Luxembourg, France, Belgium), collecting data over such a long period was a real challenge. Astronomy is an outdoor activity that is subject to the vagaries of the weather, so we had to be available and ready as soon as conditions were favorable. We also had to bear in mind that the length of the observation sessions also varied greatly from one season to the next: they were shorter on summer nights, whereas winter nights were much longer. Coping with these constraints enabled us to capture a large number of images that correspond to normal observing conditions—unlike surveys carried out in conditions that are ideal for capturing astronomical data, i.e., with space telescopes or professional ground-based observatories located in deserts.

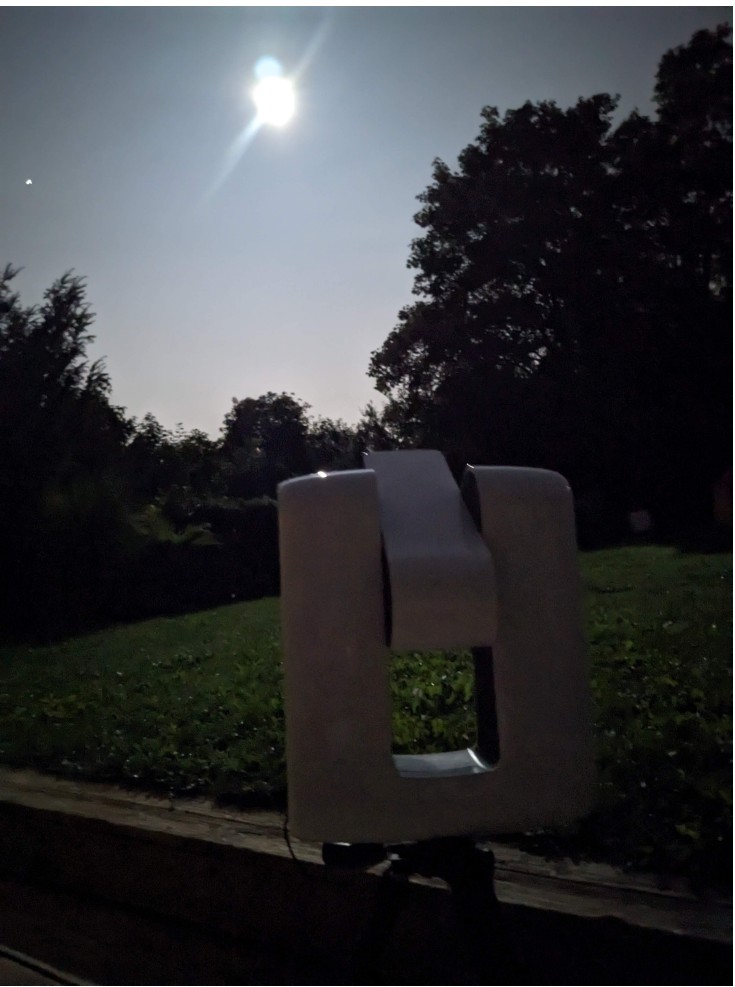

**Figure 1.** The Stellina smart instrument during an observation session in September 2023.

*4.2. Data Annotation*

Defining the bounding boxes of images was realized with a semi-automatic process. To this end, a Python prototype was developed to annotate images by relying on Python packages like openCV (4.7.0) [5] and scikit-images (0.20.0) [6], and by using StarNet (2.0.0) [7]: this deep learning model consists of removing stars from astronomical images—producing an incredible visual effect for all kinds of celestial objects. The Python prototype was executed on a Ubuntu server (20.04) with cores having 128 GB RAM (Intel(R) Xeon(R) Silver 4210 CPU @ 2.20GHz) and NVIDIA Tesla V100-PCIE-32GB.

Here, we applied the StarNet model on the RGB images: as a result, we obtained both images and their starless versions. With starless images, it is then easier to identify the DSO that are still present in the image: in fact, planetary nebulae and galaxies are generally well preserved by StarNet, while globular clusters leave their mark. Large nebulae are visible and different from the background (even if noisy).

Then, we applied classical CV algorithms on starless images to obtain the bounding box [8]. Once we had them, we mapped theses contours onto original images (i.e., with stars) (Figure 2). This computation is not perfect, especially with the faintest DSO, small apparent galaxies and large stars. To go further, we refined annotations using the Make Sense tool [15]: this interactive web-based tool allows one to draw and resize bounding boxes with a mouse, and finally store the annotations using the YOLO format (Figure 3).

As a result, we computed and refined the bounding boxes corresponding to the DSO positions—while ignoring stars. These informations were then stored in labels files (Section 3).

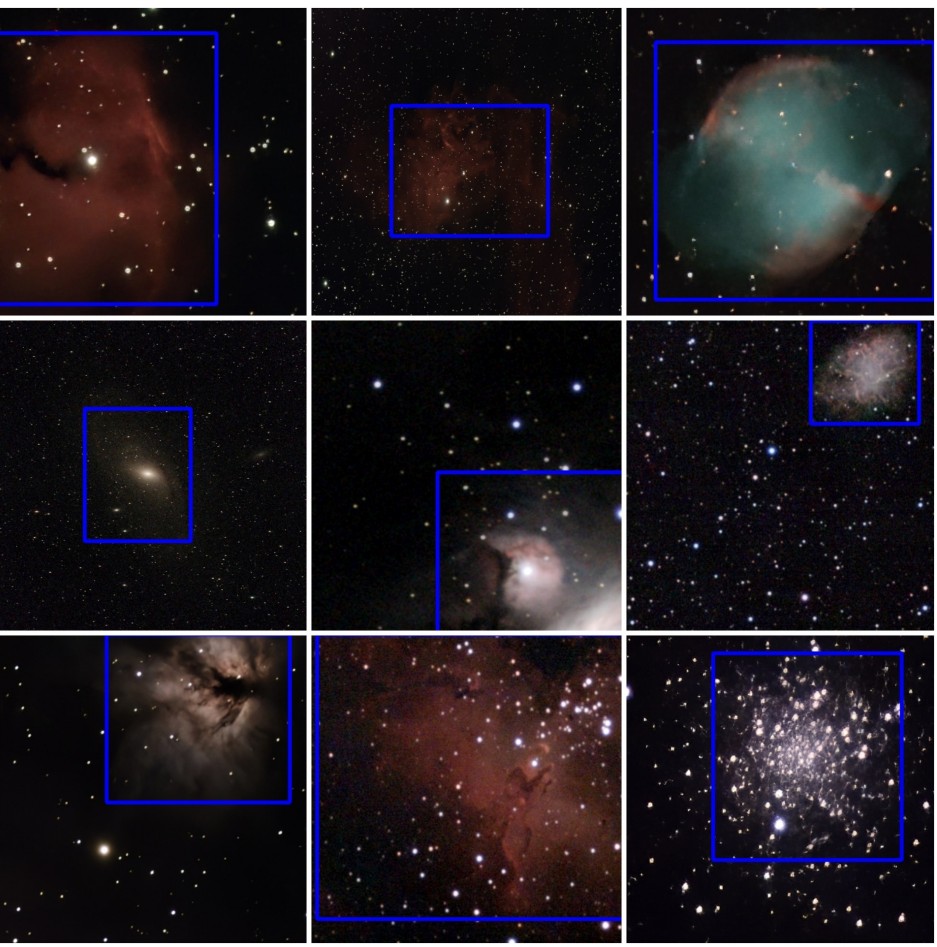

**Figure 2.** Subset of annotated astronomical images, highlighted with the bounding boxes corresponding to the labels. DSO, from top to bottom and left to right: IC2177, IC405, M27, M31, M42, M1, NGC2024, M16, M12. Boxes correspond to the contours of DSO.

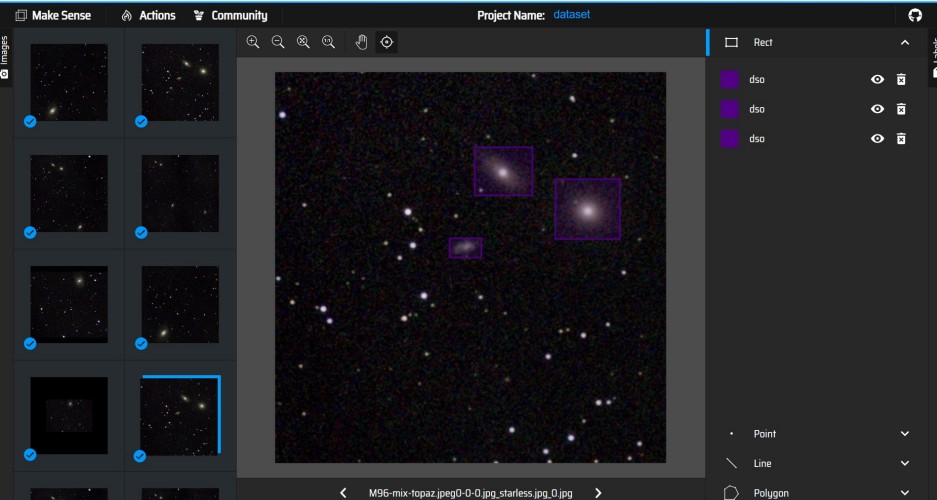

**Figure 3.** The Make Sense web application was used to refine the bounding boxes, especially with small apparent galaxies. This software allows one to export the results into the YOLO format.

## 5. User Notes

DeepSpaceYoloDataset can be used directly to train a deep learning model. For example, we trained a single-class YOLOv7 model on it (without additional data) by

applying transfer learning (i.e., using the default pre-trained YOLOv7 model [9]). To this end, the official YOLOv7 implementation was used [14] with the following parameters:

```
python3 train.py --weights yolov7.pt
                 --data ''data/custom.yaml''
                 --single-cls
                 --workers 8 --batch-size 4 --img 608
                 --cfg cfg/training/yolov7.yaml
                 --name yolov7-all
                 --hyp data/hyp.scratch.p5.yaml
                 --epochs 50
```

With the same hardware as described in (Section 4), we obtained a YOLOv7 model with an acceptable accuracy (Figure 4), leading to a model that was able to detect the presence and positions of DSO in RGB astronomical images (see Supplementary Materials to watch videos produced with the model).

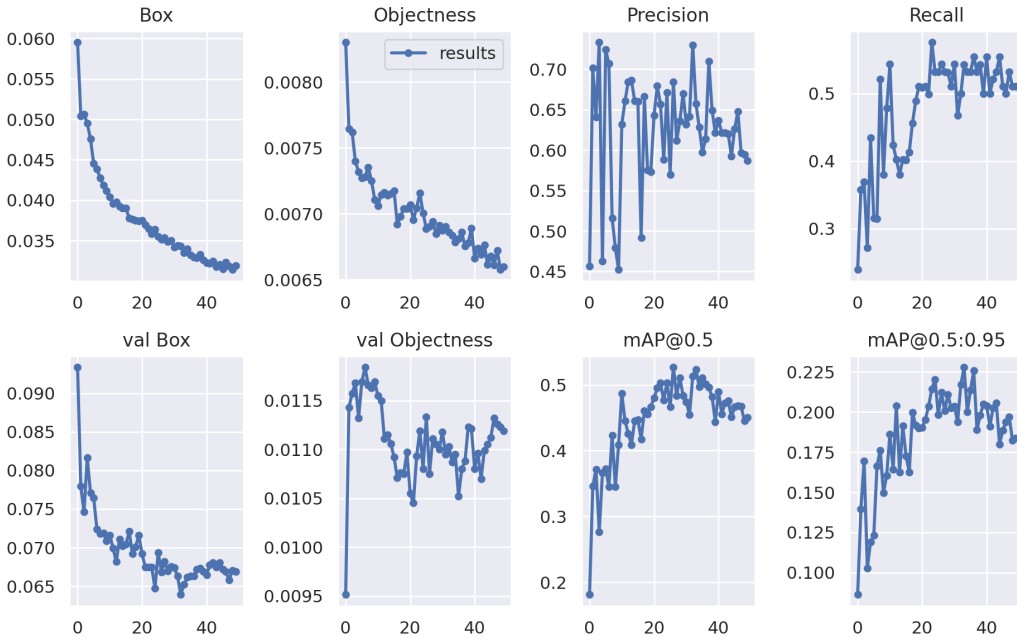

**Figure 4.** Training of a YOLOv7 model by using DeepSpaceYoloDataset: 80% of images were used as training set, 10% of images were used as test set, and 10% of images were used as validation set.

From DeepSpaceYoloDataset, it is now possible to obtain a more efficient model using fine-tuning (architecture and hyperparameters), data preprocessing (for instance, by making different classifiers dedicated to small and large objects), and even by designing a dedicated deep learning pipeline, as recently shown in [11]. The dataset can be directly used to train a model with the recent YOLOv8 [18].

## 6. Conclusions and Perspectives

This paper describes DeepSpaceYoloDataset, a set of 4696 annotated astronomical RGB images under the YOLO format. DeepSpaceYoloDataset was prepared after a long period of data collection (15 months), and then annotated by applying a semi-automatic process supported by the StarNet Deep Learning model, computer-aided contour drawing and manual refinement with dedicated software (Makesense).

Researchers, industrials, and even enthusiast amateurs can use this dataset to design, train, and apply AI models from images with noise and impacted by light pollution.

In future works, we will prepare new releases of the dataset by collecting and annotating additional images (especially from the Southern Hemisphere) and by enriching contextual information about DSO.

**Supplementary Materials:** The following supporting videos can be visualized here: 'AI-powered observation of galaxies in Leo Triplet (M65, M66, NGC3628)' https://youtu.be/C56gxz0AwoQ?si=MohQz8eOEp8DEVtq and 'AI-powered observation of Leo Group 1 galaxies (M95, M96)' https://youtu.be/R15a9Y3s1Ec?si=uR1bcoWtbvGgaqFC (accessed on 1 December 2023).

**Funding:** This research was funded by the Luxembourg National Research Fund (FNR), grant reference 15872557.

**Institutional Review Board Statement:** Not applicable.

**Informed Consent Statement:** Not applicable.

**Data Availability Statement:** Data can be found here: https://zenodo.org/record/8387071 (accessed on 1 December 2023). Additional materials used to support the findings of this study are available from the corresponding author upon request.

**Acknowledgments:** Data processing and model training were realized on the LIST Artificial Intelligence and Data Analytics platform (AIDA) https://www.list.lu/en/institute/rd-infrastructures/data-analytics-platform/(accessed on 1 December 2023), thanks to Raynald Jadoul and Jean-Francois Merche.

**Conflicts of Interest:** The author declares no conflicts of interest.

## Abbreviations

The following abbreviations are used in this manuscript:

| | |
|---|---|
| EAA | Electronically Assisted Astronomy |
| AI | Artificial Intelligence |
| DSO | Deep Sky Objects |
| CV | Computer Vision |
| YOLO | You Only Look Once |

## Notes

1. https://github.com/jiangyx123/SSOD-dataset (accessed on 1 December 2023).
2. https://pixinsight.com/doc/docs/MARS/MARS.html (accessed on 1 December 2023).
3. https://vaonis.com/stellina (accessed on 1 December 2023).
4. https://vaonis.com/vespera (accessed on 1 December 2023).
5. https://pypi.org/project/opencv-python/ (accessed on 1 December 2023).
6. https://pypi.org/project/scikit-image/ (accessed on 1 December 2023).
7. https://www.starnetastro.com (accessed on 1 December 2023).
8. https://docs.opencv.org/4.x/da/d0c/tutorial_bounding_rects_circles.html (accessed on 1 December 2023).
9. https://github.com/WongKinYiu/yolov7/releases/download/v0.1/yolov7.pt (accessed on 1 December 2023).

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
