# Peer review of "DeepSpaceYoloDataset: Annotated Astronomical Images Captured with Smart Telescopes"

_data, 2023_

Round 1

Reviewer 1 Report

Comments and Suggestions for Authors

1) Abstract should be more extensive.

2) Which software did you use for annotation that are compatible to YOLO? It should be mentioned in the manuscript. Lines 62-63. 

3) Why did you choose Luxembourg, France and Belgium for your dataset? Lines 78-79.

4) Lines 137-140: Transfer learning should be more detailed.

Author Response

Dear Reviewer,

Thank you for your comments.

I followed your recommendations to improve the manuscript, and I have updated the abstract, introduction and conclusion sections. 
Furthermore, I have addded more details about related works, and I have added technical informations about the implementation of the semi-automated process.
I also removed the 'Image Quality Assessment' short section because it is not appropriate for this paper.

I hope these changes will convince you that the manuscript is valuable for the MDPI Data journal.

Kind regards,

Olivier Parisot

Reviewer 2 Report

Comments and Suggestions for Authors The paper describes a dataset with basic analysis and missing information. The initial experiments made on this dataset are basic ones without analysis of different, commonly known and used tools. I suggest rejection of it. My main concerns are:
reviewed in this stage:
1. The abstract is very short, without any details about proposed dataset, also no information about the need for such a dataset. 
2. The introduction is short and there is no problem formulation for such a dataset. Also, the authors should analyze the latest solution in analyzing video/images and dataset state in this area. The proper introduction should explain why such a dataset is needed and where it can find applications.
3. The related works are based on 4 references, where the authors used Yolo which is not the latest version. Moreover, there are many newer learning transfer models. Proposing a new dataset should be tested on commonly known tools to show that there is a need for such a dataset and the creation of new classification tools.
4. Sec. 4 is without any details on the parameters of hardware, no information about differences
in datasets, etc. 5.  Is it balanced datasets?  
6. Fig. 2 does not show the proper analysis of datasets, more statistical analysis should be performed. Also, the charts should be prepared in better quality and axes names.
7. Sec. 5 shows a basic test, without any analysis of different tools, etc. 
8. Conclusion in general is not here. Two sentences are given without any information, conclusions, etc.

Author Response

Dear Reviewer,

Thank you for your comments.

I followed your recommendations to improve the manuscript and to explain the approach, and I have updated the abstract, introduction and conclusion sections. 
Furthermore, I have addded more details about related works, I added recent references about YOLO-based techniques and I have added technical informations (both hardware and software) about the implementation of the semi-automated process .
I also removed the 'Image Quality Assessment' short section because it is not appropriate for this paper.

I hope these changes will convince you that the manuscript is valuable for the MDPI Data journal.

Kind regards,

The author

Round 2

Reviewer 2 Report

Comments and Suggestions for Authors

The paper was improved according to my comments.